# Population-scale dietary interests during the COVID-19 pandemic

Kristina Gligorić [1]✉, Arnaud Chiolero[2,3,4], Emre Kıcıman [5], Ryen W. White [5] & Robert West [1]

The SARS-CoV-2 virus has altered people's lives around the world. Here we document population-wide shifts in dietary interests in 18 countries in 2020, as revealed through time series of Google search volumes. We find that during the first wave of the COVID-19 pandemic there was an overall surge in food interest, larger and longer-lasting than the surge during typical end-of-year holidays in Western countries. The shock of decreased mobility manifested as a drastic increase in interest in consuming food at home and a corresponding decrease in consuming food outside of home. The largest (up to threefold) increases occurred for calorie-dense carbohydrate-based foods such as pastries, bakery products, bread, and pies. The observed shifts in dietary interests have the potential to globally affect food consumption and health outcomes. These findings can inform governmental and organizational decisions regarding measures to mitigate the effects of the COVID-19 pandemic on diet and nutrition.

[1] School of Computer and Communication Sciences, EPFL, Lausanne, Switzerland. [2] Population Health Laboratory (#PopHealthLab), University of Fribourg, Fribourg, Switzerland. [3] Institute of Primary Health Care (BIHAM), University of Bern, Bern, Switzerland. [4] School of Population and Global Health, McGill University, Montreal, Canada. [5] Microsoft Research, Redmond, Washington, United States. ✉email: kristina.gligoric@epfl.ch

The coronavirus disease 2019 (COVID-19) pandemic has led to the implementation of unprecedented non-pharmaceutical interventions, including case isolation, social and physical distancing measures, business and school closures, travel restrictions, and full-scale national lockdowns[1]. For instance, in mid-May 2020, more than one-third of the global population was under lockdown[2]. These interventions have caused important shifts in people's lives, which in turn created challenges that did not originate directly in the virus itself, but in the social, economic, and psychological implications of the population-scale measures taken to prevent the spread of the virus[3,4], transforming education[5], exercise habits[6], mental health[7], online behaviors[8,9], labor markets[10], transport, and mobility[3,11], to name a few. Identifying how the pandemic has broadly impacted human needs and interests[12,13] is therefore critical.

A thorough understanding of changes in food-related interests is particularly pressing, as changes in diet can have important ramifications for health, and dietary monitoring can help improve the well-being of populations. Diets are suspected to have become less balanced during the COVID-19 pandemic[14], and changes in diet and physical activity during the pandemic are known to increase the risk of cardiovascular disease[15] and are suspected to be associated with negative mood during lockdowns[16]. The implemented interventions disparately impact population segments within a country, depending on people's demographics, health, and habits[17–19]. Therefore, the pandemic can negatively impact the diet especially of those populations and individuals who are already most vulnerable[20], such as those affected by malnutrition[21,22], eating disorders[22,23], addictions[24], or obesity[25,26]. Furthermore, in general, diet and nutrition are prominent factors in maintaining overall health and are important for developing a healthy immune response, which affects the speed of recovery and the probability of developing severe symptoms[27].

Public health and nutrition researchers and stakeholders have therefore issued a number of warnings about the potential nutritional public health issues that might emerge as a consequence, such as alcohol misuse[24] or weight-gain and obesity[18,28]. There are concerns about the long-term implications of stress and boredom associated with lockdowns, as well as emotional eating[29–31], potentially linked with alcohol misuse and weight-gain[24,28]. However, it is not clear which aspects of the many potential adverse impacts of confinement on diets are most pressing, and on which of the many potential public health issues to focus first.

Beyond health, the question of COVID-19-induced shifts in dietary interests is also of economic importance[32]. It is necessary to understand emerging consumer needs, subsequent market readjustments[33,34], and supply chain issues[35] that impact global access to food and food security[36,37]. Many emerging customer behaviors are of interest to retailers and business owners during lockdowns, such as stockpiling, more frequent cooking, online purchasing, and changes in shopping locations[17,38–42].

Early on in the course of the pandemic, anecdotal reports about changes in dietary habits during lockdowns emerged, e.g., about an increased interest in baking[43,44]. Existing research has studied the impact of COVID-19 stay-at-home orders on health behaviors and physical and mental health[45], finding initial evidence of increased sedentary behaviors and reduced physical activity[19,46–48], less eating out, increased cooking and baking from scratch[49–51], and generally increased consumption of[52,53], and interest in[54], food. Overall, food consumption and meal patterns were mostly found to be more unhealthy during confinement[51,55], with the exception of a decrease in alcohol consumption[19,55].

Current evidence, however, relies primarily on surveys and does not leverage passively collected large-scale observational data[41,56–58]. It remains challenging to quantify shifts in food interests globally and holistically, across different types of food, and fundamental questions about food interests during the pandemic remain unanswered.

The present study aims to bridge this gap by asking the following guiding question: How did dietary interests shift during COVID-19-induced mobility restrictions in 2020? To address this question, we quantify people's change of interest in foods when they spend more time at home and how long these shifts in interests persist as mobility reverts to normal. The fact that—unlike most previous events that directly impacted so many lives worldwide—the COVID-19 pandemic unfolded in a time of widespread Internet access allows us to conduct a population-wide infodemiology[59] study by relying on passively sensed digital trace data. Specifically, we use time series capturing the popularity of Google search queries related to 1,432 foods (e.g., "bread", "pizza"), as well as ways of accessing food (e.g., "recipe", "restaurant", illustrated in Fig. 1), obtained in aggregated form via the publicly available Google Trends tool, to analyze changes in food-related interests across 18 countries. Google is the world's largest Web search engine, and Google Trends search volumes have been shown to be a powerful population-scale sensor for numerous human behaviors, including unemployment[60], trading decisions[61], and voting[62]. We thus add to a rich literature that, well before COVID-19, has begun to analyze health and nutrition behaviors using digital trace data[63,64], such as search engine logs[65,66], purchase logs[67–69], online recipes[70], reviewing platforms[71], social media such as Twitter[72–75] or Instagram[76,77], and geo-location signals[78].

Methodologically, drawing meaningful conclusions from the longitudinal Google search volume time series is challenging due to the presence of trends and seasonalities. We overcome these hurdles via quasi-experimental time-series analyses (outlined in Fig. 2), isolating the effect of the 2020 discontinuity in mobility patterns on food interests and going beyond simple correlations by accounting for 2019 baseline trends. This study design lets us identify the immediate, short-term increases in interest in all food types, which is found to be stronger and longer-lasting than those that coincide with end-of-year holidays (Fig. 3a). The increased food interest is not uniform across types of food. The most prominent increases, in absolute and relative terms, occur for calorie-dense carbohydrate-based foods such as pastries and bread. The identified shifts in interests, many of which persisted for months and some of which continued past our observation period (Supplementary Fig. 12), represent a potential danger for public health and should be taken into account to inform decisions made by stakeholders in efforts to mitigate the effects of the COVID-19 pandemic on diets worldwide.

## Results

We curated a set of 1,432 entities related to specific foods (e.g., "bread", "pizza") grouped in 28 food categories (details in Methods, Fig. 2), which covered 95.7% of the global food search volume in 2019 and 2020. Supplementary Table 1 summarizes the descriptions of food categories and contains examples of popular foods in each category. We also curated a set of 16 different entities related to ways of accessing food (e.g., "recipe", "restaurant"), grouped in four categories: entities can be related to consuming food at home or outside of the home, and orthogonally, entities can be related to consuming food prepared by persons from within the household or food prepared by a third party (Supplementary Data 1). As a first glimpse into the food entities and fluctuating interests, we estimate interest in recipes

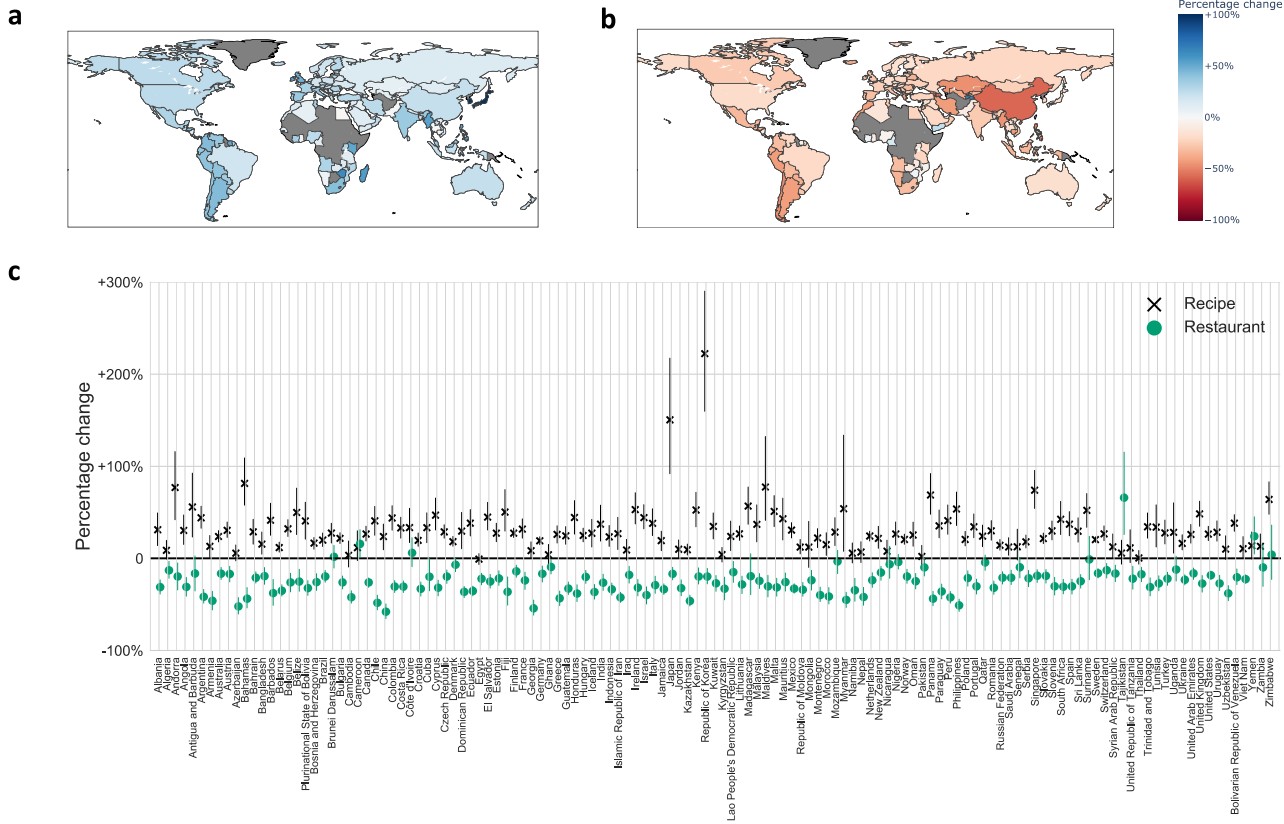

**Fig. 1 Global shifts in dietary interests.** In **a** search interest in the concept of "recipe" across countries, in 2020 vs. 2019. In **b** search interest in the concept of "restaurant" across countries, in 2020 vs. 2019. In (**a**) and (**b**), color marks average relative change in interest in 52 weeks of 2020 in a country, compared to corresponding weeks of 2019. Countries with not enough search data are marked in gray. In **c** average relative change in interest in 52 weeks of 2020 in a country, compared to corresponding weeks of 2019. Points mark average relative change (across $n = 52$ weeks), illustrated on maps in (**a**) and (**b**). Error bars mark 95% confidence intervals. During 2020, compared to 2019, there was a global increase in interest in recipes and a global decrease in interest in restaurants.

and restaurants, globally across 129 countries with enough search data available to estimate weekly interest volumes. Figure 1 illustrates the emerging shifts: a global increase in interest in recipes (Fig. 1a), and a global decrease in interest in restaurants (Fig. 1b) during the weeks of 2020, compared to the corresponding weeks of 2019.

We collected search interest time series in 18 countries. The studied countries were selected to achieve geographic diversity and comprise countries with a large number of Internet users across continents: Brazil, Canada, Mexico, United States, France, Germany, Italy, Spain, United Kingdom, India, Indonesia, Japan, Egypt, Kenya, Nigeria, and Australia. Additionally, to achieve a varying severity of lockdowns, Sweden and Denmark were added as contrasting cases, due to particularly lenient COVID-19-induced restrictions[79]. The interest time series were collected from the Google Trends platform[60,80] and calibrated with Google Trends Anchor Bank[81] (so time series for different search queries can be aggregated via summation and compared with one another). Although absolute search volume—the number of issued queries—cannot be inferred, calibration can infer absolute search volume up to a constant multiplicative factor. This way, ratios of absolute search volumes can be validly estimated when working with calibrated Google Trends time series. The interest time series in the same regions in 2019 serve as baselines.

Note that, although different languages are spoken in the 18 studied countries, search queries did not need to be translated, as Google Trends allows language-independent entity descriptors from the Freebase knowledge base[82] as input. For instance, given

as input the Freebase entity descriptor for "bread" (/m/09728), Google Trends will return the search interest for all queries related to the concept "bread" across languages.

**Overall surge in food interest larger than during end-of-year holidays.** We examine how the total interest in food entities evolved in 2019 and 2020 (Fig. 3a). We monitor interest in all food entities, normalized by the 2019 mean and standard deviation ($z$-scores). We refer to this quantity as the surplus of interest. Normalizing food interest allows us to quantify the surplus of food interest in a week relative to the Christmas week of 2019. In a given week, the surplus relative to the Christmas week is measured as the ratio between the $z$-score in the observed week and the $z$-score in the Christmas week.

First, note the peaks of food interest during the end-of-year holiday season in both 2019 and 2020. Second, note the increase in overall interest in food entities coinciding with the reduced mobility due to COVID-19 occurring in March 2020. These rises of food interest are larger in amplitude compared to the rises of interest during end-of-year holidays, and they last longer. For example, in the US, the surplus (compared to the 2019 mean) of food interest at its peak during the first wave of the COVID-19 pandemic equals the surplus of interest during the Christmas week of 2019, as well as that of the surplus of interest during the Thanksgiving week of 2019. In total, the surplus of food interest in the first six months of 2020 in the US is 9.0 times higher than the surplus of interest during the Christmas week of 2019, and 8.9

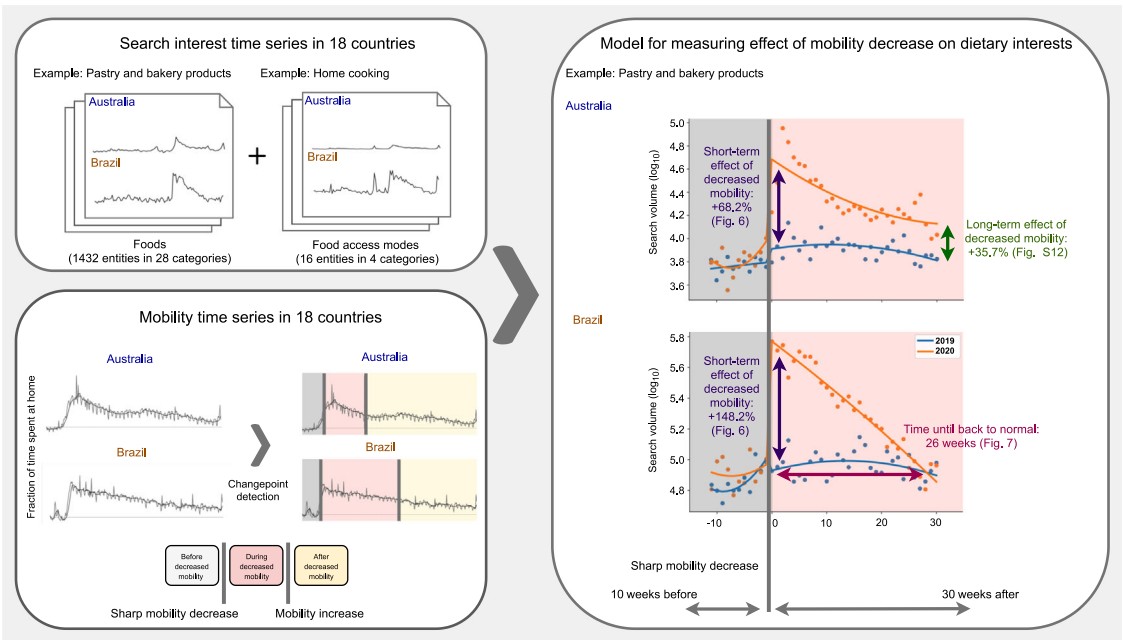

**Fig. 2 Study design.** We start from interest time series in 18 countries, capturing search interest in food entities and in entities about ways of accessing food. In order to measure the effect of the changes in mobility time series on interest, we first detect mobility changepoints (the abrupt mobility decrease and the eventual mobility increase) via changepoint detection. On the right, we illustrate the modeling approach on the example of interest in pastry and bakery products in Australia and Brazil, where on the x-axis is the week relative to the week of the mobility decrease, and on the y-axis is search interest. The modeling approach measures the effect of the shock of mobility decrease on dietary interests, controlling for pre-pandemic trends. With this model, we measure three key quantities: the short-term effect of decreased mobility, the time until interest reverts back to normal, and the long-term effect of increased mobility.

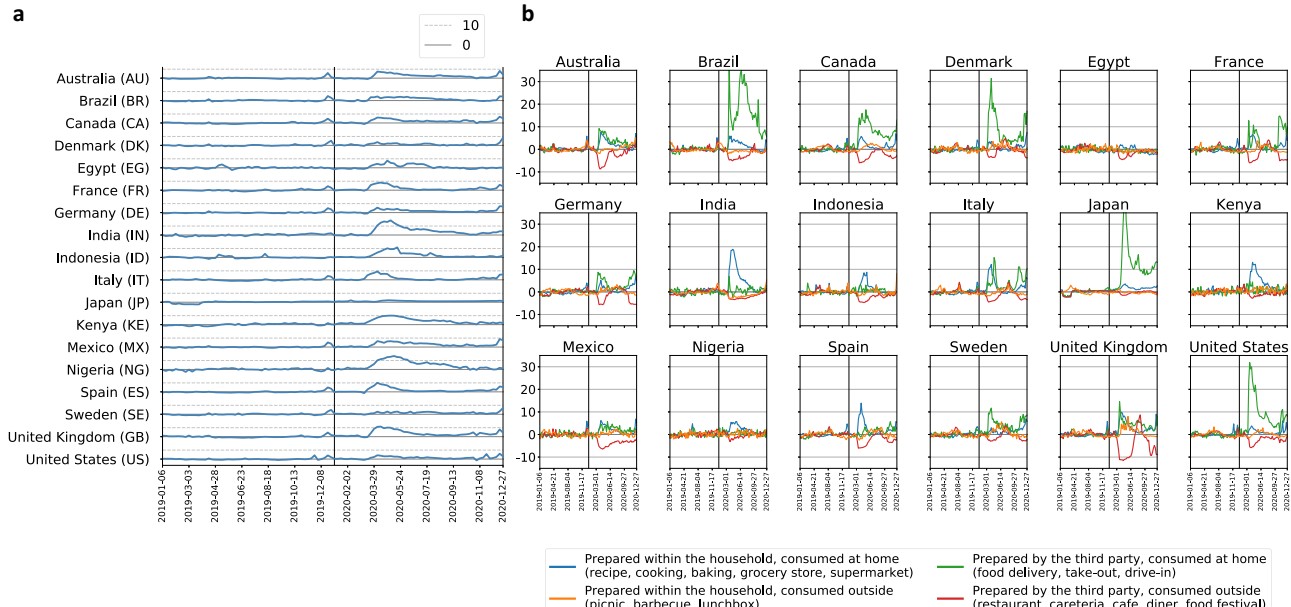

**Fig. 3 Temporal evolution of dietary interests.** In **a** total interest in food entities (z-scores), 2019–2020. The dashed line marks ten standard deviations above the 2019 mean. In **b** interest in ways of accessing food (z-scores), 2019–2020.

times higher than the surplus of interest during the Thanksgiving week of 2019.

We next compare the surplus of food interest at its peak during the first wave of the COVID-19 pandemic with the surplus of interest during the Christmas week of 2019 across countries. We exclude countries with a non-Christian majority (India, Indonesia, Japan, Egypt, and Nigeria), where there are no prominent increases in food interest during Christmas week (Fig. 3a). When

comparing to Christmas holidays, the surplus of food interest at its peak during the first wave of the COVID-19 pandemic is on average 1.9 times higher than the surplus of interest during Christmas week, while the total surplus of interest in the first six months of 2020 is on average 18.8 higher than the surplus of food interest in the Christmas week of 2019.

The increases in food interest are drastic in India, Indonesia, and Nigeria, too, with food interest at the peak of mobility

restrictions surpassing ten pre-pandemic standard deviations. Note that Sweden, Denmark, and Japan, the countries with the mildest government-mandated mobility restrictions[79,83–85], contrary to all other studied countries, had no notable overall increase in food interest in 2020 (Fig. 3a).

Next, similarly, in Fig. 3b, we examine the temporal evolution of the interest in the four modes of accessing food reflecting whether they relate to consuming food at home or outside of the home, and orthogonally, whether they are related to consuming food prepared by persons within the household or food prepared by a third party. In all countries, in 2020, there was a decrease in interest in food prepared by third parties and consumed outside (in red), and an increase in interest in food prepared within the household and consumed at home (in blue) coinciding with the onset of the first wave of the COVID-19 pandemic in the first half of 2020.

Compared to the end-of-year holidays, the surplus of interest in food prepared within the household and consumed at home (recipes, cooking, baking, grocery stores, and supermarkets) was at the peak 1.7 times higher than the surplus during the Christmas week of 2019, on average across countries (excluding countries with a non-Christian majority; cf. above). In the first six months of 2020, it was in total 13.7 times higher than the surplus of interest during the Christmas week of 2019. The increases in interest in recipes, cooking, baking, grocery stores, and supermarkets relative to the 2019 mean were large in India as well, surpassing ten pre-pandemic standard deviations at the peak. Additionally, we note large increases in interest in food prepared by third parties and consumed at home (in green). In the United States, Brazil, Japan, and Denmark, this interest increased by more than 30 pre-pandemic standard deviations at the peak.

**Changes in food interests are strongly associated with mobility**. Next, we combine search interest time series with mobility data published by Google (described in Methods) which captures the relative increase in time people spend indoors compared to a pre-pandemic baseline. We find that interest in different ways of accessing foods and interest in specific foods are strongly correlated with mobility during the COVID-19 crisis (Fig. 4a, b).

Across weeks in 2020, from February to the end of December 2020 (the period for which mobility data is available), we calculate the country-specific Spearman rank correlation between mobility time series and food interest time series. Here, in order to adjust for seasonal trends, the food interest for a given week of 2020 is expressed as a relative increase compared to the corresponding week of 2019.

We observe strong and significant associations between food interests and mobility. Interest in recipes (Fig. 4b) is positively correlated with spending more time at home ($p < 0.05$ in all countries except Japan; Spearman's rank correlation coefficient ranging between 0.36 in Egypt and 0.95 in Mexico), and takeout is significantly and positively correlated in 13 out of the 18 studied countries (Spearman's rank correlation coefficient ranging between 0.34 in Indonesia and 0.82 in Australia). Interest in restaurants, on the other hand, is negatively correlated with spending more time at home, significantly in all studied countries (Spearman's rank correlation coefficient ranging between −0.40 in Kenya and −0.97 in Italy).

Regarding food categories (Fig. 4a), despite some variation between countries, there are notable food categories that have a significant positive correlation with spending more time at home in most of the studied countries, such as desserts (ranging between 0.40 in Sweden and 0.84 in Brazil) and pastries and bakery products (ranging between 0.46 in Denmark and 0.88 in the United Kingdom).

The correlation between mobility and food interest normalized by the 2019 baseline is measured for individual entities (Supplementary Data 1). All entities related to consuming food at home are correlated positively on average over countries, whereas all entities related to consuming food outside of the home are correlated negatively on average (except barbecue, likely due to the fact that barbecue food can also be consumed at home). Among specific foods, the strongest positive correlation is found for pancake, baking powder, bread, baker's yeast, cookie, chocolate brownie, chicken meat, chocolate cake, biscuit, and pasta. The strongest negative (although much smaller) correlation is found for foods such as tapas, Korean barbecue, sushi, and gelato, typically eaten in social contexts taking place outside of the home.

In the analyses so far, we have examined the response of the interest as the mobility changed by measuring correlation. Next, given the abrupt nature of the change in mobility, we isolate the effect of the shock of mobility decrease on food interest via a modeling approach.

**More interest in home food, less interest in out-of-home food**. As depicted in Fig. 2, to isolate the shock of the mobility decrease occurring in all studied countries in March 2020, we first automatically detect changes in the mobility time series caused by government-mandated lockdowns or self-motivated social distancing measures (Methods). We refer to these points as mobility changepoints (Supplementary Fig. 1).

To measure the effect of decreased mobility on food interest time series, we employ a quasi-experimental design that isolates the impact of the mobility decrease shock (the discontinuity), controlling for patterns occurring in the same weeks of 2019 when COVID-19-induced mobility restrictions did not occur (Fig. 2). The model of a given interest time series in a given country is given by the following regression discontinuity design (RDD) in quadratic form:

$$
\begin{aligned}
\log y_{tT} = \ & \alpha' && + \beta' \cdot t && + \gamma' \cdot t^2 \\
& + \alpha'' \cdot i_t && + \beta'' \cdot i_t t && + \gamma'' \cdot i_t t^2 \\
& + \alpha''' \cdot j_T && + \beta''' \cdot j_T t && + \gamma''' \cdot j_T t^2 \\
& + \alpha \cdot i_t j_T && + \beta \cdot i_t j_T t && + \gamma \cdot i_t j_T t^2,
\end{aligned}
\tag{1}
$$

where $T$ is the year (2019 or 2020); $t$ is the week in the year relative to the week in which the discontinuity occurred in 2020 (but not in 2019), for $t \in [-t_{\min}, t_{\max}]$; $t_{\min} = 10$, since it is the maximum number of weeks in 2020 before the cutoff; $t_{\max} = 30$, since it is the maximum number of weeks we can have so that across all the studied countries, the second mobility decrease shock is not included; $y_{tT}$ is the calibrated (see above) search interest volume in week $t$ of year $T$ of an entity (or set of entities) in the respective country; $i_t$ is a binary variable equal to 1 if $t > 0$ and 0 otherwise; and $j_T$ is 1 in 2020 and 0 in 2019. This way, for all weeks where $i_t = j_T = 1$, a unit is "treated", otherwise it is not. Logarithmic outcomes are used in order to make the model multiplicative. The outcome is modeled as a separate quadratic function of time before and after the discontinuity in order to capture nonlinear temporal patterns. By comparing observations lying closely on either side of the temporal threshold, we estimate the treatment effect while minimizing potential bias from unobservable confounders.

The interaction coefficients $\alpha$, $\beta$, $\gamma$ model the effect of the discontinuity, controlling for baseline trends in 2019. The short-term increase in interest is captured by the fitted coefficient $\alpha$, which estimates the short-term effect of the mobility decrease on search interest. The approach is described in more detail in Methods and outlined in Fig. 2.

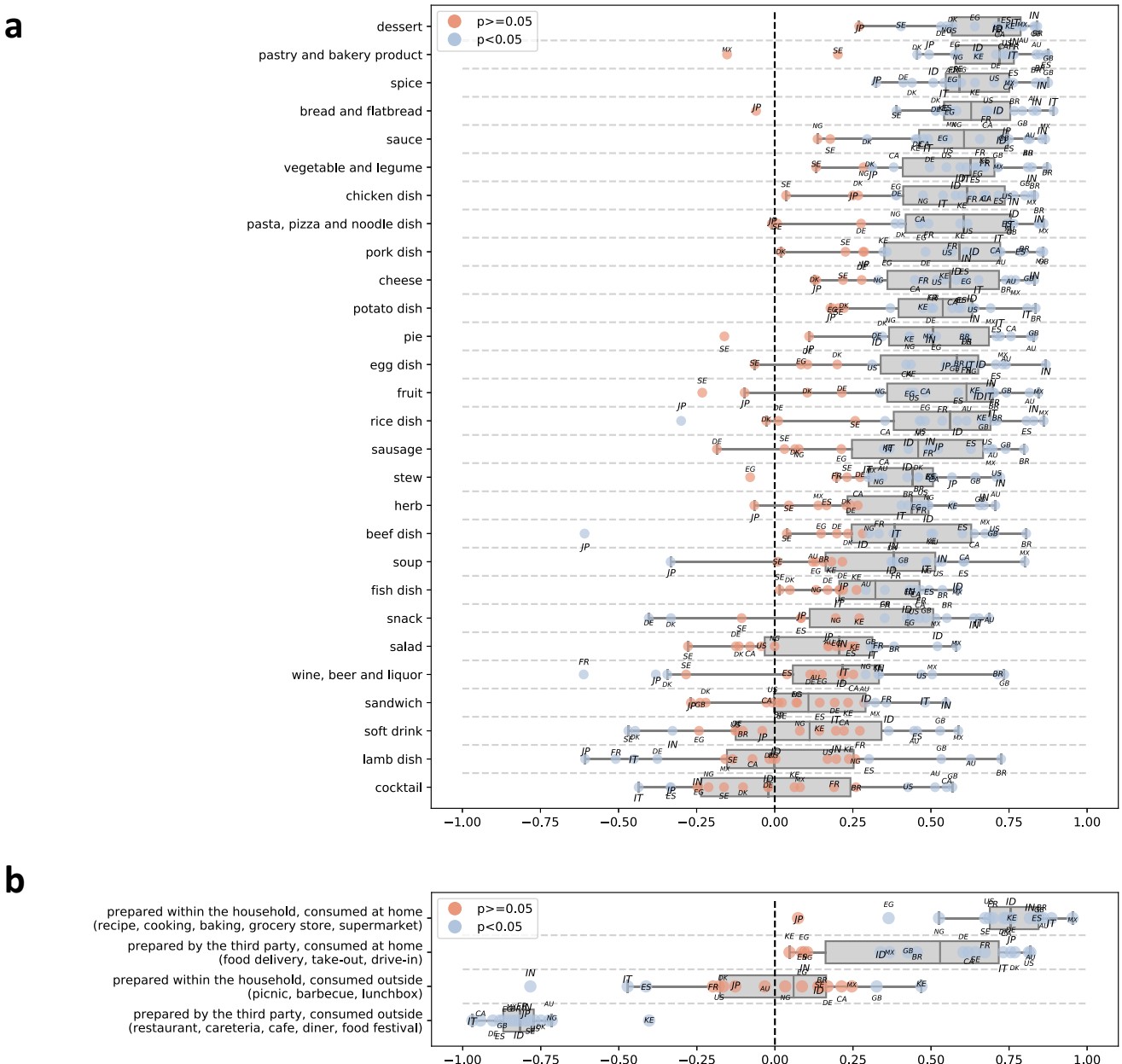

**Fig. 4 Spearman rank correlation coefficient between mobility decrease and interest volume.** In **a** correlation for categories of food entities, and in **b** for ways of accessing food. For each group, $n = 18$ values represent the correlation coefficients for the 18 studied countries (calculated based on $n = 46$ samples corresponding to weeks of 2020). The boxplot summarizes the values across 18 countries. Significant correlations ($p < 0.05$), according to a two-sided t-test (with no correlation between interest and mobility as the null hypothesis) are marked in blue, and non-significant correlations in orange. No adjustments for multiple comparisons are made. Boxplots represent the 50th (centerline), 25th, and 75th percentile (box limits). The whiskers extend to the minimum and maximum values but no further than 1.5 times IQR.

We find that in 15 out of the 18 studied countries (Fig. 5a), there was a significant short-term increase in interest in food prepared within the household and consumed at home (e.g., recipes), with a short-term boost in interest ($\alpha$) ranging between +32.2% in Egypt and +179.8% in India. In 14 countries, there was a significantly decreased interest in food prepared by third parties and consumed outside of the home (e.g., restaurants), ranging between −32.1% in the USA and −81.7% in France. There were major increases in interest in food prepared by third parties and consumed at home (e.g., food delivery), with more than a +100% significant increase in eight of the 18 studied countries.

We next analyze the relationship between the amplitude of the short-term changes in dietary interest and the severity of lockdowns (Fig. 5b), where the severity of a lockdown is defined as the percentage change of the fraction of time spent at home (with respect to the pre-pandemic baseline level) at the peak of reduced mobility[86]. The severity of a lockdown varies between +10.1% in Sweden, the country with no government-mandated mobility restrictions[84], and +31.6% in Italy, a country with severe lockdown measures[87]. All peaks of mobility decrease occurred between March and May 2020.

We find that the more drastic the lockdown severity, the more drastic the change in dietary interests. Changes in interest in

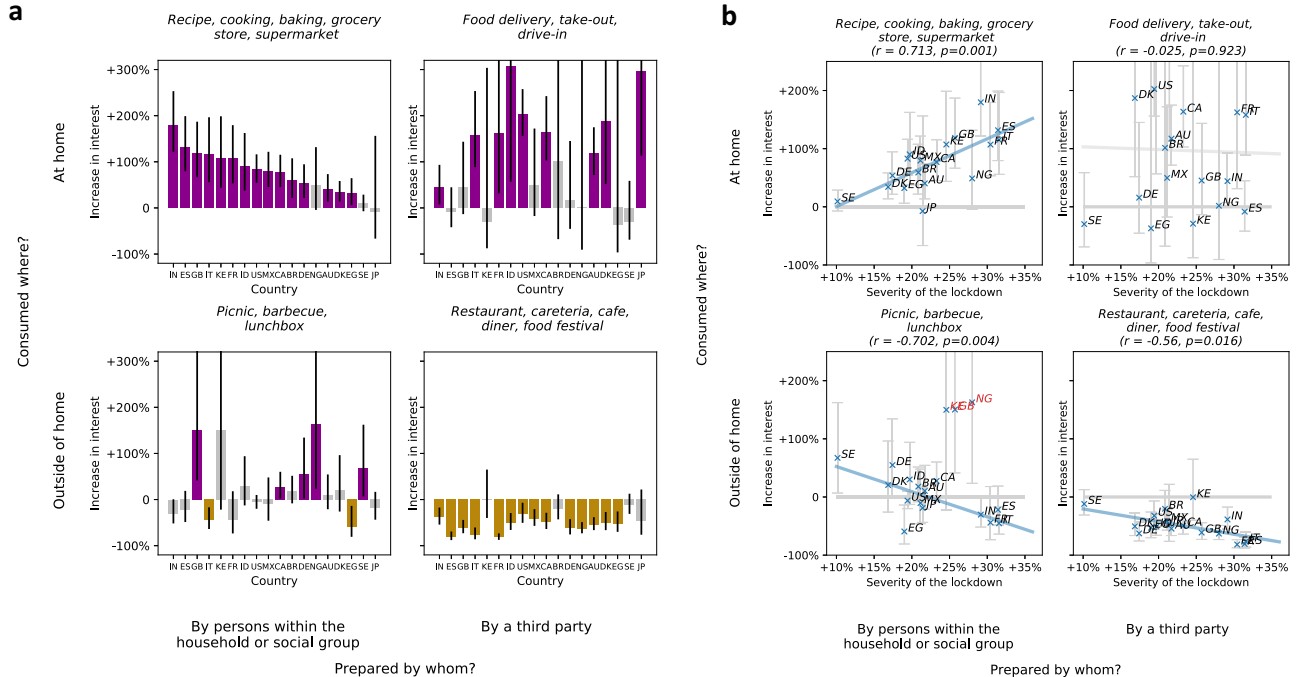

**Fig. 5 The short-term effect of mobility decrease on interest in accessing food.** In **a** the short-term effect of the shock of mobility decrease on interest in accessing food, estimated with our RDD-based model. For each group of entities (n = 4), and each country (n = 18), the model (Eq. (1)) is fitted on n = 82 samples representing weekly interests. Bars represent effect estimates (fitted coefficient α). Error bars mark 95% confidence intervals. Purple marks significant positive (p < 0.05), yellow significant negative (p < 0.05), and gray marks nonsignificant effects according to a two-sided t-test. Fitted coefficients and statistics are presented in Supplementary Table 2. In **b** the relationship between the severity of the lockdown, measured as the increase in the percentage of time spent at home at the peak of reduced mobility (x-axis) and the estimated short-term effect on interest (y-axis), across four groups of entities about ways of accessing food. For each way of accessing food (n = 4), Pearson correlation coefficient is reported (calculated over n = 18 countries) and p-value according to a two-sided t-test with no correlation between the severity of lockdown and short-term effect as the null hypothesis. The short-term effect of the shock of mobility decrease on interest is estimated with our RDD-based model and presented with 95% confidence intervals. The straight blue line is a least-square fit. Red country codes mark countries that are considered outliers. No adjustments for multiple comparisons are made.

food-access modes have an association with the severity of the lockdown: positive for food prepared within the household and consumed at home (R = 0.71, p = 0.001), and negative for consumption outside of the home, i.e., food prepared by third parties and consumed outside of the home (R = −0.56, p = 0.016) and food prepared by third parties and consumed at home (R = −0.70, p = 0.004). Here, the United Kingdom, Kenya, and Nigeria were excluded because they are clear outliers. When not excluding these three countries, we still observe a negative, but nonsignificant correlation (R = −0.11, p = 0.676). The discrepancy between the United Kingdom, Kenya, Nigeria and the other countries might be linked to COVID-19 policies allowing congregation in open green spaces, including parks and beaches[88]. Note that Sweden, the country with no government-mandated mobility restrictions[84], had no significant shift of interests in ways of accessing food.

The fact that the effect of decreased mobility on the interest in recipes and similar entities across countries rises linearly with the severity of lockdown adds to the evidence that interests changed because mobility decreased. If there were other confounding factors that could explain the changes in dietary interests, and those factors had nothing to do with the shock of the mobility decrease, we would not expect to find such a clear dose-response relationship. Instead, we would need to envisage a more complex effect of an unobserved factor that could impact both the strength of the lockdown in a country and cause changes in the population's dietary interests, in ways that have nothing to do with spending more time at home.

Although significant increases in interest in food prepared by third parties and consumed outside of the home also exist, they are not correlated with lockdown strength. Presumably, other factors are at play, such as the response of the market, availability of delivery companies, or how quickly restaurants adapted to do deliveries.

**Drastic increases in interest for calorie-dense, carbohydrate-based foods**. Having established the link between the sudden decrease in mobility and the shifting interests in ways of accessing food, we next examine how exactly the interest in specific types of food varied (Fig. 6). Is the observed increase in food interest uniform across all food types, with interest in all foods increasing proportionally, or does interest in certain foods increase more? We apply the modeling approach (Fig. 2) on time series capturing interest in the 28 food categories in the 18 countries and measure the short-term effect of decreased mobility.

Overall, we find that there was a momentary increase of total food interest (gray bands in Supplementary Fig. 10), significant in all countries except Sweden and Japan, ranging between +24.6% in Denmark and +99.4% in Spain. Similarly, there was an increase in interest in most of the individual food categories (Fig. 6). The biggest increases, however, occurred for calorie-dense, processed, carbohydrate-based foods: pastry and bakery products, bread and flatbread, and pie. These effects are significant in most countries. Especially strong cases (with increases of over 200%) include pastry and bakery products in Spain, France, Canada, and Egypt; bread and flatbread in Spain, France, and Italy; and pie in Spain.

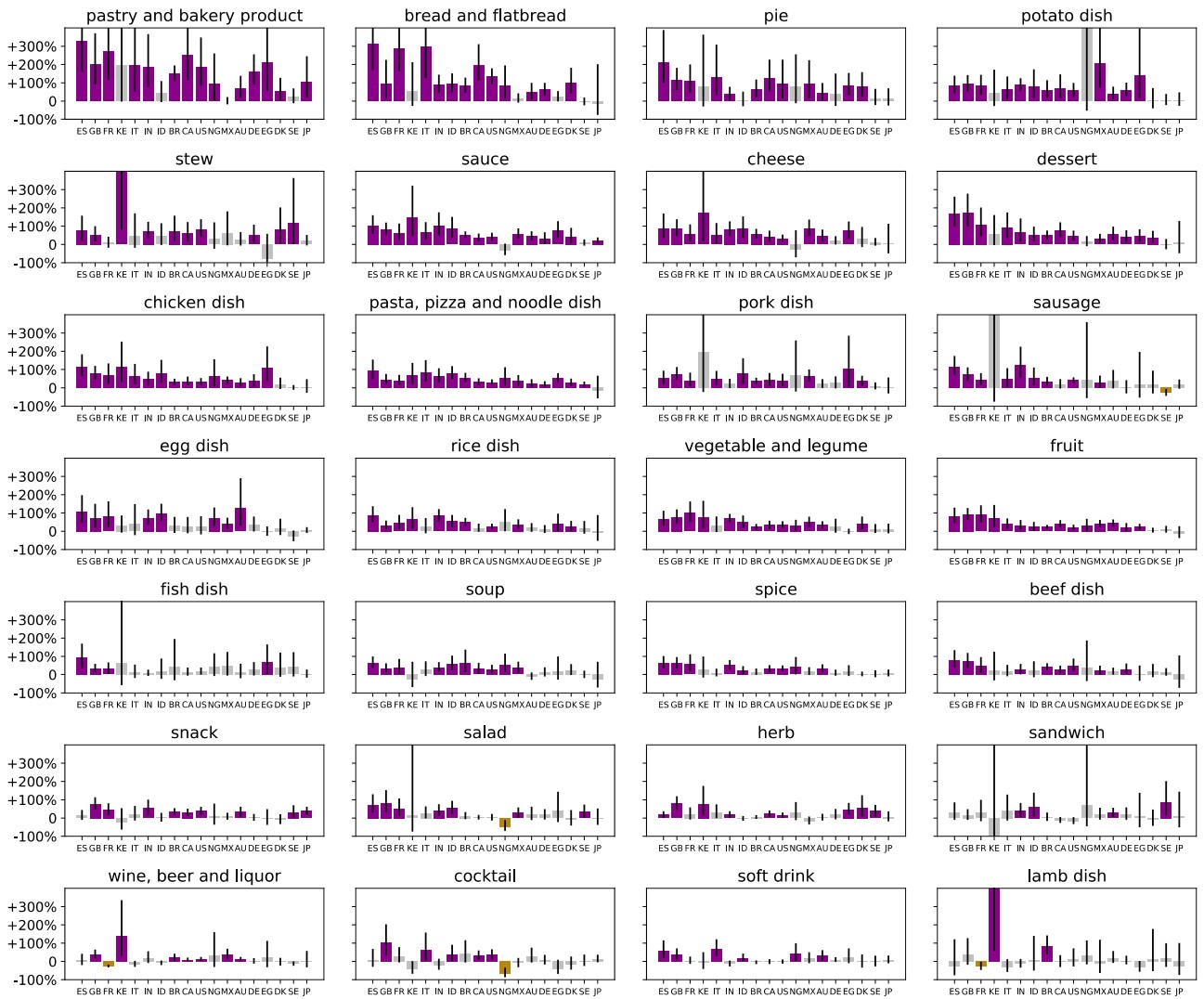

**Fig. 6 The short-term effect of the shock of mobility decrease on interest in categories of food entities.** The effect is estimated with our RDD-based model. For each category of entities ($n = 28$), and each country ($n = 18$), the model (Eq. (1)) is fitted on $n = 82$ samples representing weekly interests. Bars represent effect estimates (fitted coefficient $\alpha$). Error bars mark 95% confidence intervals. Purple marks significant positive ($p < 0.05$), yellow significant negative ($p < 0.05$), and gray marks nonsignificant effects according to a two-sided $t$-test. Fitted coefficients and statistics are presented in Supplementary Table 2. Food categories are sorted by median effect across countries. No adjustments for multiple comparisons are made.

We observe smaller increases for other categories, including fresh produce (vegetable, fruit, salad, herb), meat and fish dishes (chicken, pork, fish, beef, lamb dishes), and wine, beer, liquor, and cocktails, which saw an increase in some of the countries. These conclusions and the relative ranking between categories are robust to specific modeling choices (Supplementary Material and Supplementary Table 3).

In the Supplementary Material (Supplementary Fig. 9), we additionally provide an alternative analysis where the outcome variable is the relative volume share (i.e., the fraction of the total weekly food interest that is allocated to the respective search queries), rather than absolute volume as analyzed above. This way, we control for the overall increased food interest. In terms of the share of interest, the most prominent increases again occurred for pastry and bakery products (over 50% increase in share fraction in 11 of the 18 countries) and bread and flatbread (over 50% increase in share faction in six of the 18 countries), whereas the share of interest in other food categories remained robust or decreased slightly. The most prominent decreases in terms of the share of interest occurred for soft drinks, alcoholic drinks, and sandwiches—beverages and food presumably

typically consumed in social contexts taking place outside of the home.

Although most food categories saw increased interest in most countries, there are specific foods where interest decreased as mobility decreased, such as tapas and energy drinks (Supplementary Data 1).

We next measure the time it took for search interest to revert to normal, illustrated in Fig. 2 for the example of Brazil. We measure how many weeks after the mobility decrease it takes until the modeled interest in 2020 is no longer significantly different from the counterfactual prediction based on 2019 (based on non-overlapping 95% confidence intervals). In addition to being large in amplitude, we observe that the numerous shifts in interests lasted for months. For instance, the shortest duration of increased interest in food prepared within the household and consumed at home was 6 weeks (in Egypt; Fig. 7a), and the shortest duration of increased interest in specific food categories, 9 weeks (wine, beer, and liquor in France, pastry and bakery products in Denmark, and sandwiches in Sweden; Fig. 7b). Most of the changes in interest in specific groups of food entities are transient, and the interest returned to normal within 30 weeks.

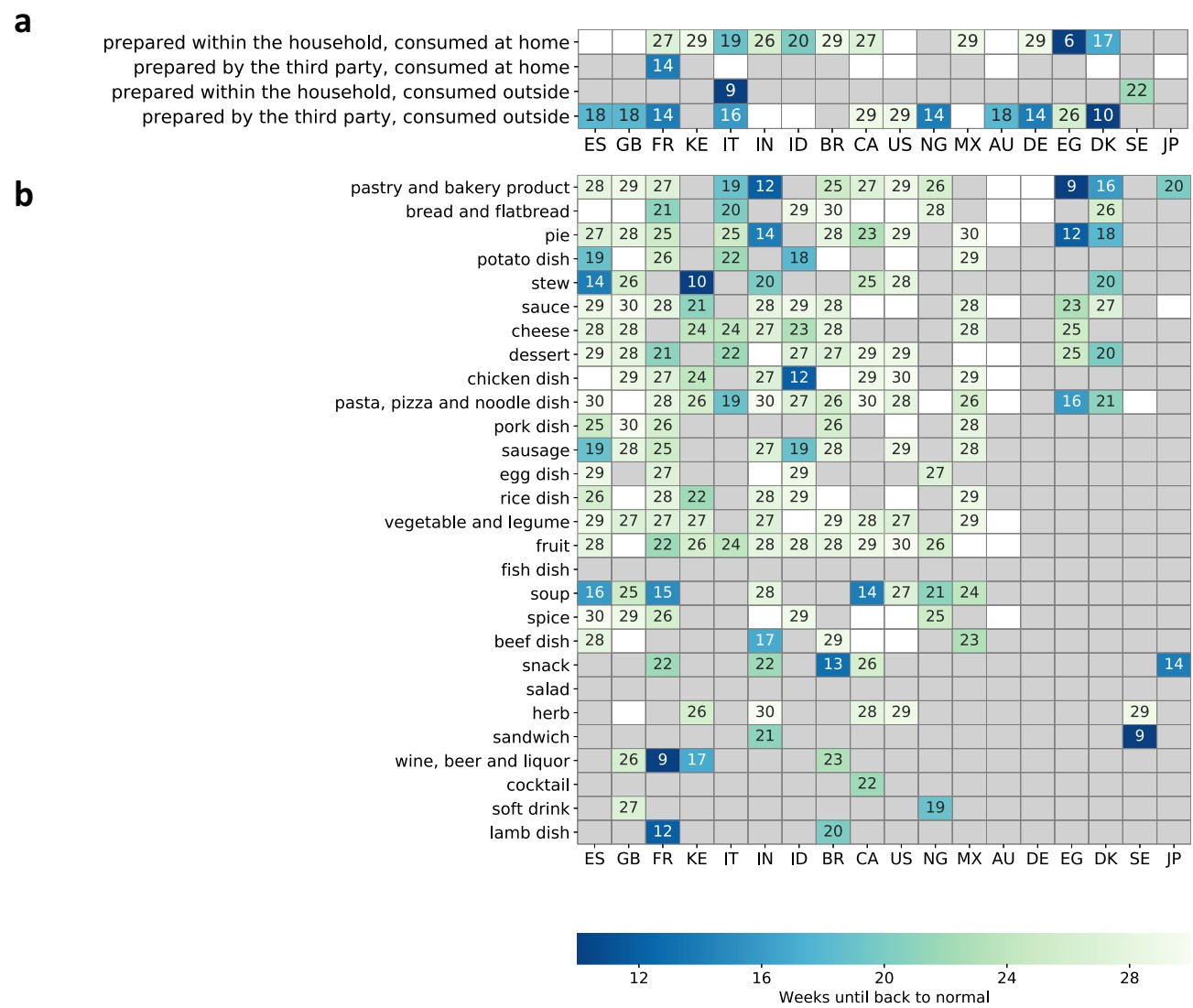

**Fig. 7 Number of weeks until food interest goes back to normal.** In **a** the number of weeks for ways of accessing food, and in **b** for food categories. The number of weeks is determined by measuring how many weeks after the mobility decrease, the modeled interest in 2020 is no longer significantly different from the counterfactual prediction based on 2019 interest, based on non-overlapping 95% CI. Gray marks that there were no significant differences in 2020 compared to 2019, and white marks that the interests did not go back to normal until the end of 2020.

In cases where interest did not return to normal within the 30 weeks after the mobility decrease, we measure (in Supplementary Fig. 12) how elevated the interest remains at the end of the modeled period, 30 weeks after the mobility decrease, compared to the interest in the same week in 2019 (illustrated in Fig. 2 using the example of Australia). While most interests come back to normal within 30 weeks, there are some notable exceptions of more permanent changes (Supplementary Fig. 12): the interest in food prepared by third parties and consumed at home (e.g., food delivery) permanently increased in Italy, Canada, the United States, Australia, Denmark, and Japan, while the interest in food prepared within the household and consumed at home (e.g., recipes) also remained elevated in Spain, the UK, the US, and Australia.

**Second wave had less impact on food interests**. Finally, we explore the effect of the "second wave" of the pandemic, which occurred between October and December 2020 in the UK, Canada, Italy, the US, France, and Spain. In Supplementary Fig. 11a, we observe much smaller effects in the second wave compared to the first wave of mobility decrease. No significant increases in interest in food prepared within the household and consumed at home (e.g., recipes) are observed. While mobility saw large changes in some countries (Supplementary Fig. 1), no drastic changes in food interests occurred.

Notable exceptions are France and Italy, where food prepared by a third party and consumed at home saw large increases in interest in the second wave. We observe significant decreases in interest in restaurants in the UK, Italy, and France, but smaller effects compared to the first wave. Finally, no notable surges in interest in bread, pastry, baking, and desserts as in the first wave are observed in the second wave. The second wave brought on less drastic mobility decreases and was less of a disruption. Additionally, as populations adapt and acquire new skills, there might be less of a need for recipe searching over time.

## Discussion

In order to formulate policies and allocate resources for mitigating the adverse nutritional impacts of the COVID-19 pandemic, governments, organizations, non-governmental

organizations, and other stakeholders need reliable and timely data regarding the circumstances faced by affected populations. The results presented here document the impacts of confinement on nutritional interests. As the pandemic continues to unfold, warnings about potential public health issues emerge. In this study, we aim to point out prominent emerging behaviors by quantifying initial developments and providing a broad grounding for future studies.

From a public-health perspective, the emerging surge in food interest during confinement is concerning. During the first wave of the COVID-19 pandemic, there was an overall surge in food interest, stronger and longer-lasting compared to the end-of-year holiday season of 2019 (Fig. 3a). Since Christmas and Thanksgiving are known to be disruptive to dietary habits and a hazard to balanced diets[89], and the effects of confinement on food interests are comparable in amplitude and last longer, there is a pressing need to understand them.

In addition to the overall volume, the nature of the food interest also changed. After the shock of the mobility decrease, there was a large immediate increase in interest in consuming food at home and a decrease in consuming food outside of the home. In 12 of the 18 studied countries, as mobility decreased, the interest in preparing and consuming food at home momentarily increased by more than 50% (Fig. 5a), and the interest in baking and pastries more than doubled (Fig. 6). Since such modified interests persisted for a prolonged period (at least 9 weeks, Fig. 7) and since frequent consumption of meals prepared away from home is significantly associated with an increased risk of all-cause mortality[90], preparing more meals at home is a potentially positive side of the shifts in interest and should be understood further from a public-health perspective.

However, the sharply increased interest in potentially calorie-dense foods is worrisome. Overall, we find that the most drastic increases in interest are in carbohydrate-rich foods. These surges are not matched by proportional increases in interest in fresh produce, meat meals, vegetables, or fruit. Such shifts represent a danger of favoring processed and calorie-dense foods, at times when physical activity is reduced. This is particularly concerning from a population-scale well-being and mental-health point of view. These results call for developing a deeper understanding of the exact mechanisms in how stress, boredom, and emotional eating associated with the lockdown may have contributed to the observed effects[29–31]. It is also necessary to understand further how changes in product availability (e.g., a shortage of fresh ingredients including milk, eggs, and flour[91] in many countries) and changes in consumer behaviors (including the emergence of stockpiling[39]) are linked with the increases in interest in carbohydrate-rich foods.

Supplementary Fig. 12 hints at permanent small increases in interest in certain foods. While the interest in restaurants came back to normal in the studied countries except India, Indonesia, and Mexico within 30 weeks after the shock of the mobility decrease, interest in takeout remained increased in Italy, Canada, the United States, Australia, Denmark, and Japan, and interest in recipes in Spain, the UK, the United States, and Australia also remained increased.

Future work should determine if these are new permanent habits brought on by the pandemic, or if they will fall back to normal in the future. These findings are particularly important to take into account in efforts to understand market readjustments.

Our results confirm and refine what is known from survey-based research. A meta-analysis[55] of 12 preliminary articles studying the impact of COVID-19 confinement on dietary habits revealed a sharp rise in carbohydrate consumption, especially of foods with a high glycemic index (e.g., homemade pizza, bread, cake, and pastries), as well as more frequent snacking. High consumption of fruits and vegetables, as well as protein sources, particularly pulses, was also recorded, although there was no clear peak of increase in the latter. A decrease in alcohol intake and of fresh fish and seafood was further observed.

Whereas surveys are potentially a more accurate reflection of consumption, our findings, which were derived from passively sensed data, provide a complementary view. Search interest time series capture fine-grained temporal dynamics within the contrasting periods. Additionally, search interest time series are not subject to the reporting biases of surveys. By relying on them, we account for behavioral changes beyond subjective impressions. Finally, search interest time series provide insights at a population scale.

Contrary to previous concerns about the danger of alcohol abuse during confinement[24] triggered by stress, boredom, and emotional consumption, on a population level, we do not observe important surges in interest in alcohol consistent with these concerns. In fact, consistent with survey-based research[19,55], we observe a significant negative correlation between seasonality-adjusted interest in alcoholic drinks and mobility in some of the studied countries (cocktails: $-0.44$ [$p = 0.002$] in Italy, $-0.33$ [$p = 0.02$] in Japan, $-0.33$ [$p = 0.02$] in Spain; wine, beer, and liquor: $-0.61$ [$p = 6 \times 10^{-6}$] in France, $-0.38$ [$p = 0.009$] in Japan, $-0.34$ [$p = 0.02$] in Denmark), meaning that more interest in alcohol is associated with more time spent outside of the home, not less. Additionally, the relative share of interest in alcoholic drinks (Supplementary Fig. 9) decreased since the increase in other foods was not mirrored by the increase of interest in alcoholic drinks.

It is important to remember that these findings are based on aggregate population-level interests and that a specific sub-population of users might still be susceptible to alcohol misuse. Future work should study search logs and alternative digital traces[92] of individual users in a longitudinal user-level study to understand what pre-pandemic user characteristics are predictive of behaviors emerging during confinement.

When interpreting our results, several additional considerations should be kept in mind. First, searching for food is not tantamount to consuming the food. Users may search but not consume, and vice versa. Also, search interest might not be an equally good sensor for real behavior in different countries. Note, however, that several factors nonetheless render our findings consequential. In other contexts, digital traces of nutritional behavior have been shown to be valid proxies of actual behavior[72]. Additionally, even if traces are imperfect proxies, major shifts in search interest have the potential to impact actual food consumption. In that sense, search interest, one of the few global signals that are publicly accessible to researchers and policymakers, can lead to consumption.

Second, while we make no claims of causal identification based on our statistical analyses, our regression discontinuity-based design alleviates the effect of unobserved covariates by exploiting the sudden shock in mobility and accounting for seasonal variation. The observed dose-response relationship (Fig. 5b) supports this, as does the fact that search interest in ways of accessing foods behaves as one would expect if those interests were causally affected by mobility.

Third, the data collection capacities limited the number of studied countries such that interest data could feasibly be collected. Our results may not be representative beyond the 18 countries studied here. Still, given the globally shifting interests in ways of accessing food (Fig. 1), we believe the results from the countries studied here are indicative of shifts in interests in neighboring countries.

Finally, beyond people's shifting habits, interests, and emotional responses, other internal and external factors brought by

the pandemic, most notably food product availability, price, and expected shelf life[91] or populations' present level of cooking skill and willingness and ability to learn to cook[93] can play a role and should be kept in mind when interpreting the observed shifts in dietary interests.

Outside of the ongoing COVID-19 pandemic, spending more time at home due to enforced lockdowns is a naturally occurring implicit dietary intervention encouraging people to eat at home. By documenting the impacts on people's interests and measuring how lasting the effects are we learn something about the kinds of foods that people become interested in when staying at home in general. This has implications for designing interventions outside of COVID-19, and future work should compare effects on a diet of staying at home due to COVID-19 lockdown measures to the impacts of staying at home due to other, more frequent external circumstances, such as extreme weather or air pollution.

We study and document the impacts of a single event (COVID-19 crisis), but we observe similar impacts across culturally and geographically different countries. The observed impacts are therefore general to a certain extent, applying to different kinds of populations, in varying intensity depending on the intensity of the treatment.

When confined, people are interested in carbohydrates and calorie-dense foods (Fig. 6), likely due to changes in preferences[29,30] on the one hand and due to changes in accessibility and price of foods[91] on the other hand. These effects are consistent across countries, which is a demonstration that they occur across cultures and economic conditions.

While this study quantified initial developments during the pandemic, future studies aiming to understand the impacts of the pandemic and the related mobility restrictions on diet will continue to be important for designing policies and programs to tackle adverse health impacts.

## Methods

**Search interest time series**. Our analyses rely on a curated and calibrated set of interest time series collected from Google Trends, an important tool for researchers[60,80] that makes aggregate statistics about the popularity of search queries in the Google search engine publicly available. We collect time series of search interest in entities related to foods or ways of accessing foods. Search queries may be specified as plain text (e.g., "Cookie") or as entity identifiers (e.g., "/m/ 021mn") from the Freebase knowledge base[82]. We use Freebase identifiers to conduct a multilingual study of interest since they allow for grouping various surface forms relating to the same topic. For instance, the entity "Cookie" ("/m/ 021mn") captures "cookies", "cookie", "Cookie", or "cookie jar", etc., while the entity "Recipe" ("/m/0p57p") captures all recipe queries across languages.

Google Trends provides time series of search interest for the specified input queries. Since search interest is not returned in terms of absolute search volume, but normalized by time and location and rounded to integer precision, we use Google Trends Anchor Bank (G-TAB)[81] to calibrate the time series. The benefit of calibration is that the interest is expressed on the same scale and the combined interest in a set of entities can be estimated by adding up the interest in individual entities.

We collect interest data for two types of Freebase food entities: (1) entities related to the ways how people access food (such as "recipe", "restaurant") and (2) specific food entities (such as "cookie", "pizza").

1. Food-access mode entities: we curate entities that reflect ways of accessing food, starting from seed entities (recipe, take-out, restaurant, picnic), and inspecting related entities. Food-access mode entities are aggregated into four groups. Entities can be related to consuming food at home or outside of the home; orthogonally, entities can be related to consuming food prepared by persons within the household or food prepared by a third party (see Supplementary Data 1 for details about individual entities). We refer to the four groups of entities related to food:

    (a) prepared within the household, consumed at home: recipe, cooking, baking, grocery store, supermarket
    (b) prepared by the third party, consumed at home: food delivery, take-out, drive-in
    (c) prepared within the household, consumed outside: picnic, barbecue, lunchbox
    (d) prepared by the third party, consumed outside: restaurant, cafeteria, cafe, diner, food festival

2. Food entities: we start from ids of food entities from Freebase. These are entities of type "food", "dish", "beverage", or "ingredient". Food entities are aggregated into categories. Category creation: we enrich Freebase entities with Wikidata knowledge base[94] properties using the Wikidata query API. For each Freebase entity id, we query Wikidata with the Freebase id to get its "instance of" or "subclass of" properties. We derive a taxonomy of 28 categories based on "subclass of" and "instance of" relations. To ensure that the food classes are general and representative, we keep all classes with at least ten entities. Note that not all entities have a "subclass of" or "instance of" field available in Wikidata and therefore cannot be automatically categorized. To achieve higher coverage, we manually annotate a set of popular entities. We monitor the global time series of all food entities in 2019–2020. We select the top entities that covered 95.7% of global food search volume and annotate all such entities that do not already have a category derived based on Wikidata. This process resulted in a set of $N = 1432$ entities, categorized either based on Wikidata or manually. Categories are presented in Supplementary Table 1. An author who is a professional epidemiologist specializing in nutrition assessed and refined the entities and the corresponding categorization.

**Search interest data collection**. Overall, we collected time series for 1432 food entities and 16 food-access mode entities in 18 countries, spanning from January 1, 2019, to December 31, 2020, at weekly granularity. The time series were collected and calibrated with the Google Trends Anchor Bank library. The full list of 1432 food entities and 16 food-access mode entities is available in our data repository. The 1432 food entities are categorized into 28 food categories, and the 16 food-access mode entities are categorized into four groups. After data collection, we obtain country-specific time series for 28 food categories, and four aggregate food-access modes by adding up time series of respective individual entities.

The countries were preselected with the goal of achieving global coverage across continents, studying countries with a large number of Internet users[95], and including countries with varying severity of mobility restrictions. Additionally, we collected global time series of interest in the "recipe" and "restaurant" entities, spanning from the beginning of 2019 until the end of 2020, at weekly granularity, in all countries and territories (Fig. 1).

**Mobility time series and COVID-19-induced mobility decreases**. To capture variation in the mobility of the populations in the 18 studied countries, we use mobility reports[96] published by Google, which capture population-wide movement patterns based on cellphone location signals. We use country-wise mobility data from February to the end of December 2020. The mobility reports specify, for each day, by what percentage the time spent in residential areas differed from a pre-pandemic baseline period in early 2020.

We chose to rely on mobility data and not the official start of lockdown dates. The problem with employing the official start of lockdown date in statistical analyses is that it is not guaranteed that they would impact movement patterns across different countries homogeneously (e.g., it could be that for some of the countries people stayed more at home even before the lockdown was enacted). Similarly, the official lockdown date might vary within a country.

We automatically detect changes in the mobility time series caused by both government-mandated lockdowns as well as self-motivated social distancing measures[86]. We refer to these points as mobility changepoints. We use mobility changepoints as heuristic dates for when people started or stopped spending substantially more time in their homes. Unlike choosing one of the official dates of lockdown implementation or relaxation, this leads to a meaningful onset of decreased mobility across different countries.

Supplementary Fig. 1 depicts three important mobility changepoints dates that occur at different moments throughout 2020 in the studied countries:

1. The first sharp mobility decrease occurred in March and April 2020 when people started to spend substantially more time at home.
2. The mobility increase occurred as people stopped spending substantially more time at home.
3. The second mobility decrease occurred between October and December 2020 (occured in some of the studied countries), when people started spending substantially more time at home during the second wave of the pandemic.

We detect the three changepoints for each country independently by smoothing and thresholding: we consider the weekly rolling average mobility. We monitor the percentage of time spent at home. The first date when time spent at home increased by 10% is the start of reduced mobility in the first wave. We repeat the same to detect the onset of the second wave. In this way, the period when the percentage of time spent at home consistently stays above 10% compared to the pre-pandemic baseline (defined as pre-pandemic mobility levels by Google) is a period of decreased mobility, in the first, or in the second wave. Note that the period of decreased mobility is very short in Sweden, the country with no government-mandated mobility restrictions. Sweden is still included in our analyses as a contrasting case.

The three changepoint dates are marked in Supplementary Fig. 1 in the 18 studied countries. The first mobility decrease and the second mobility decrease (in case it occurs) serve as cutoff dates in our modeling approach. The date of the mobility increase serves to limit the possible duration of the studied period with decreased mobility.

**Modeling approach.** To estimate the potential effects of the sudden mobility changes on food interest time series, we devise a regression discontinuity design (RDD) with a local regression in time. Additionally, we incorporate a fake discontinuity separating before vs. after the cutoff date in 2019, the year before the pandemic, to account for seasonal trends. The model of a given interest time series in a given country has the general quadratic form described in Equation (1).

With this form of the model, we measure the time-dependent trends because the model is expressive enough (i.e., quadratic terms capture the temporal evolution, see illustrations in Supplementary Fig. 5). We also provide the main results with constant and linear models in the Supplementary Material.

Bandwidth choices are made in the following way: $t_{min} = 10$, since it is the maximum number of weeks in 2020 before the cutoff, $t_{max} = 30$, since it is the maximum number of weeks we can have so that across all the studied countries, the second mobility decrease shock is not included. We also investigated the impact of the choice of the bandwidth (see Supplementary Material).

The interaction coefficients $\alpha, \beta, \gamma$ model the effect of discontinuity, controlling for trends in 2019. We are primarily interested in $\alpha$, the magnitude of the initial increase at the discontinuity.

In our analyses, we fit a model of this general form (Equation (1)) to interest time series, separately for each studied entity or group of entities, in each of the studied countries. We use the modeling approach to investigate three key quantities illustrated with the example of interest in pastries and bakery products in Brazil and Australia in Fig. 2:

1. Short-term increase in interest, captured with by the fitted coefficient $\alpha$. The model is multiplicative due to the logarithm. After fitting the model (Equation (1)) with OLS, the relative increase over the baseline is then calculated by converting $\alpha$ back to the linear scale, via $e^{\alpha} - 1$; the 95% CIs (approximated with two standard errors) are also converted back to linear scale.

2. The time it takes for the interest to revert to normal. We measure how many weeks after the mobility decrease (within the $t_{max} = 30$ weeks) the modeled interest in 2020 is no longer significantly different from the counterfactual prediction based on 2019 (based on non-overlapping 95% CIs).

3. Long-term increase in interest. In case the interest did not go back to normal within the 30 weeks after the mobility decrease, we measure how elevated the interest remains at the end of the modeled period, 30 weeks after mobility decrease, compared to the interest in the same week in 2019.

Data analysis was performed using Python, version 3.8. Open-source libraries were used for data analysis (matplotlib v3.1.3, pandas v1.0.3, NumPy v1.18.1, scikit-learn v0.22.1 scipy v1.4.1 statsmodels v0.11.0).

**Reporting Summary.** Further information on research design is available in the Nature Research Reporting Summary linked to this article.

## Data availability

The data generated in this study is publicly available at https://github.com/kristinagligoric/dietary-interests-covid[97]. The release at the time of submission was archived in https://doi.org/10.5281/zenodo.5805181.

## Code availability

Code is publicly available at https://github.com/kristinagligoric/dietary-interests-covid[97]. The release at the time of submission was archived in https://doi.org/10.5281/zenodo.5805181.

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

## Acknowledgements

We thank Gorjan Popovski, Manoel Horta Ribeiro, and Maxime Peyrard for help with data collection, as well as Eric Horvitz for helpful feedback. This project was funded by the Microsoft Swiss Joint Research Center. R.W.'s lab was additionally supported by the European Union (TAILOR, grant 952215), Collaborative Research on Science and Society, and gifts from Facebook and Google.

## Author contributions

K.G. and R.W. conceptualized the study. K.G. collected and analyzed the data. A.C., E.K., R.W.W., and R.W. contributed to the study design and result interpretation. All authors wrote, edited, and revised the manuscript.

## Competing interests

The authors declare no competing interests.
