## [Peer Review File · Nature Communications]

Reviewers' Comments:

Reviewer #1:

Remarks to the Author:

The current study uses Google Trends to analyse how people's search behaviours on the different types of food change as soon as there is a drastic decrease in people's ability to move around brought about by the lockdown or stay-at-home notice. The authors find a significant discontinuous change in people's dietary preferences, with a substantial increase in the search for calorie-dense carbohydrate-based food. This finding has an important policy implication on the potential effect of the COVID-19 lockdown on people's food consumption and health outcomes.

This is an amazingly well-written paper. It was very clear to me that the authors had taken a lot of time and effort to make sure that they were able to cover most, if not all, of the issues that might arise during the reviewing process. The methodology and the execution of it were correct and well carried out. The results from the Google Trends analysis are also unique and of significance to the field of food consumption and public health. I also applaud the authors for having done so well at listing all of the limitations to the paper.

If there is one thing for me to say, in the hope that I could help improve the paper further, is that I think the authors could really explain the drastic increase in the search for calorie-dense carbohydrate-based food better. While I acknowledge that the authors have mentioned briefly about the confounding influences of product availability and food prices in line 392, I thought the authors could have also referred to the real shortage of flour, milk, and eggs that many countries experienced early on in the pandemic as a more focused explanation for the increase in the demand for baked goods as well.

More specifically, what I'm trying to say here is that the increase in the search for bread/pastry/bakery product could be presented alongside an increase in the search for flour/eggs/milk around the same time. Doing so, I believe, would further strengthen your results. Given such scarcity, it should be of no surprise why the demand for baked goods went up more than any other types of food. Also, because of scarcity again, one might also argue that people buy these baked goods to freeze them whereas they are more likely to consume other types of food straightaway, which means that the authors might be overestimating the overconsumption of calorie-dense carbohydrate-based food as well.

But on a grand scheme of things, I really liked reading the paper. I think it's a great contribution to the literature and truly believe that the paper will be well-cited in the future.

Reviewer #2:

Remarks to the Author:

General comments

This is a well written manuscript that explores the shifts and trends in the online interest of various dietary terms during the COVID-19 pandemic.

It presents a concise and well-structured introduction to the topic, while the Results are presented in a structured and detailed manner, consisting of several visualizations that add to the readability of the manuscript. The Discussion well presents the implications of the results of this study, while the authors have also included the main limitations of their work.

Major comments

1. This is an infodemiology study, so a concise mention/description of the topic in the introduction could be added.
2. My main concern is this of the not so detailed GT data collection methodology. Though it is obvious that the authors know the topic well, the presentation and reporting should be enhanced.

That is, it should be presented like a step-by-step procedure that can be replicated by anyone. For example, period, category, region, quotes of not for more than one-word keywords, etc., are details that need to be included. In order to see how to structure and report a detailed GT methodology, please see the following two papers:

a) <https://www.jmir.org/2021/7/e27044/>

b) <https://www.jmir.org/2020/8/e19611/>

(Note: I am not in the list of authors of any of these articles, nor am I in anyway requesting that you cite these documents (they are totally irrelevant with your work anyway); I am just including them in order to show how this comment could be addressed).

3. Since GT data are normalized, the retrieved time series should exactly match the respective one for which non Google data (e.g., in this case mobility data) are used. I am certain that the authors have taken this into account and have adjusted the time series accordingly, but this should be explicitly mentioned in the Methods section, i.e., the authors should include the exact time frame for which they retrieved GT data, as well as when they retrieved said data, as the time of retrieval may slightly affect the downloaded time series.

4. I understand (and support) the selection of countries that are diverse enough so that the effect can be studied from different angles, but possibly the study period for each country (or at least group of countries) should vary, since not all countries were equally affected in all waves.

Minor comments

5. I think that black should be replaced with another (lighter) color (e.g., gray) in Figure 5. Imagine the ink waste if this was printed!

6. Worldwide heat maps depicting the online interest in some (or even all, like one panel figure) of the examined terms would be a plus

Reviewer #3:

Remarks to the Author:

Population-scale dietary interests during the COVID-19 pandemic

The study presents data on how life alterations related to the first wave of the covid-19 pandemic have affected interest in different dietary items at the population level. Methodologically, the authors used food-related Google Search volumes from 12 countries in the period 2019-2020. Changes to the volume and the nature of the food-related searches were visible in all countries and could be linked to the extent of mobility restrictions in the individual countries.

This is an interesting and obviously topical paper. The results are clear and well-interpreted (for the most part) within the conditions tested. The paper is methodologically valuable as the long time window considered makes it possible to provide interesting insights into how persistent these changes are over time and how long it takes for things to "go back to normal". This is a major advance compared to the existing literature on the topic which is mostly based on self-reports providing snapshots of much shorter time ranges.

My only significant criticism to the paper would be to the geographical distribution of the included countries. The authors state (LL. 110-112) that the 12 countries were chosen to achieve a diversity of geographical location but clearly there is an overrepresentation of western countries; in particular there is only one Asian country and no African country at all. This lack of representation clearly diminishes the global relevance of the paper, especially considering that almost 80% of the world population lives in Asia and Africa. Therefore, I would strongly encourage the authors to include additional countries from those regions.

A smaller issue was with the introduction, which I found, at times, insufficiently rooted in the existing literature. For example, at LL. 70-72 the authors state that a limit of existing literature is

that it has focused on specific countries. This is not really true as there are already several international comparative studies...

(such as e.g.

Molina-Montes et al. (2021). Impact of COVID-19 confinement on eating behaviours across 16 European countries: The COVIDiet cross-national study. *Food Quality and Preference*, 93, 104231.

Ismail et al. (2021). Assessment of eating habits and lifestyle during the coronavirus 2019 pandemic in the Middle East and North Africa region: a cross-sectional study. *British Journal of Nutrition*, 126, 757-766.

Ammar et al. (2020). Effects of COVID-19 home confinement on eating behaviour and physical activity: results of the ECLB-COVID19 international online survey. *Nutrients*, 12(6), 1583. – This is already included in the reference list)

... that have already established that there are differences between countries in the way dietary habits have been affected by covid-19 and even linked these differences explicitly to lockdown severity, which is one of the main findings of this paper. These studies do not make the paper any less valuable, as the topic is still highly relevant and there are issues with previous studies (e.g., reliance on self-reported measures via surveys).

Another issue that I miss in the introduction is the topic of individual differences within countries. It is well established by now that the effect of the lockdown and other restrictive measures has been different for different segments of the population defined by (sex, age, BMI, pre-existing dietary habits, etc.). This point is not directly related to the author's study which deals with population-level data, but I think it should at least be briefly acknowledged: there is substantial literature on the topic and it is disingenuous to let readers with the impression that the effect of covid measures on people's dietary interests are only (or even primarily) related to which country they live in.

Other comments:

LL. 144-146. Should a reference be provided for this statement?

LL. 198-223. I have little background in time-series analysis so I am not able to critically evaluate the modeling approach. I hope/assume another reviewer can attend to this aspect.

LL. 220-224. The measure of lockdown severity used by the authors makes very good sense but I see no reference so I assume it is an original approach; if so, it might be relevant to check whether it correlates with more formal ways of quantifying lockdown severity, e.g. using data from the Oxford COVID-19 Government Response Tracker (OxCGRT).

LL. 331-334. This link to stress, boredom, emotional eating etc., should also be discussed upfront (i.e., in the introduction) as it is often brought up in the literature. Here, I would rather discuss how the authors' results are consistent with this view; for example Fig. 4 shows increases in interest for pastries but no increase in alcoholic beverages, so it is not clear that all results align with the "emotional eating" explanation.

LL. 369-380. This part could be shortened a bit. Also, it sounds more defensive that it needs to be. It is certainly true that searching for a food is not the same as eating food, but you can say the same about the existing literature which is largely based on surveys. A well-known fact in nutrition research is that self-reported data correlate quite poorly with actual food intake and you also have issues such as social desirability bias that may lead people to over(under) report certain items. At least this paper is based on actual behaviors so in fact I would actually regard it as having a higher validity than most existing studies on this topic.

Table 1. Two minor issues: 1) Pasta does not generally contain eggs. 2) The description of the pie category ("baked dish") should be improved.

Point-by-point response to reviewer comments

Reviewer #1

R1.1 comment: *“The current study uses Google Trends to analyse how people's search behaviours on the different types of food change as soon as there is a drastic decrease in people's ability to move around brought about by the lockdown or stay-at-home notice. The authors find a significant discontinuous change in people's dietary preferences, with a substantial increase in the search for calorie-dense carbohydrate-based food. This finding has an important policy implication on the potential effect of the COVID-19 lockdown on people's food consumption and health outcomes.*

This is an amazingly well-written paper. It was very clear to me that the authors had taken a lot of time and effort to make sure that they were able to cover most, if not all, of the issues that might arise during the reviewing process. The methodology and the execution of it were correct and well carried out. The results from the Google Trends analysis are also unique and of significance to the field of food consumption and public health. I also applaud the authors for having done so well at listing all of the limitations to the paper.

If there is one thing for me to say, in the hope that I could help improve the paper further, is that I think the authors could really explain the drastic increase in the search for calorie-dense carbohydrate-based food better. While I acknowledge that the authors have mentioned briefly about the confounding influences of product availability and food prices in line 392, I thought the authors could have also referred to the real shortage of flour, milk, and eggs that many countries experienced early on in the pandemic as a more focused explanation for the increase in the demand for baked goods as well.

More specifically, what I'm trying to say here is that the increase in the search for bread/pastry/bakery product could be presented alongside an increase in the search for flour/eggs/milk around the same time. Doing so, I believe, would further strengthen your results. Given such scarcity, it should be of no surprise why the demand for baked goods went up more than any other types of food. Also, because of scarcity again, one might also argue that people buy these baked goods to freeze them whereas they are more likely to consume other types of food straightaway, which means that the authors might be overestimating the overconsumption of calorie-dense carbohydrate-based food as well.

But on a grand scheme of things, I really liked reading the paper. I think it's a great contribution to the literature and truly believe that the paper will be well-cited in the future.”

R1.1 response: Thank you for the constructive feedback and for pointing out these important considerations related to shortages. Shortages of fresh ingredients are indeed an important part of the picture that is likely linked to the increase in interest in carbohydrate-rich foods. We have incorporated a more focused discussion about the shortage of ingredients and stockpiling. While, in the Introduction, we previously discussed stockpiling as one of the emerging behaviors, we did not connect back to that aspect in the Discussion. In the revised version, we have therefore added additional considerations in the Discussion section at the end of the paper (line 354, page 7).

Reviewer #2

R2.1 comment: *“This is a well written manuscript that explores the shifts and trends in the online interest of various dietary terms during the COVID-19 pandemic.*

It presents a concise and well-structured introduction to the topic, while the Results are presented in a structured and detailed manner, consisting of several visualizations that add to the readability of the manuscript. The Discussion well presents the implications of the results of this study, while the authors have also included the main limitations of their work.

This is an infodemiology study, so a concise mention/description of the topic in the introduction could be added.”

R2.1 response: Thank you for the encouraging comments. We agree that this is an infodemiology study. We therefore added a reference to infodemiology in the Introduction (line 83, Page 2), where we state that “the fact that the COVID-19 pandemic unfolded in a time of widespread Internet access allows us to conduct a population-wide infodemiology study by relying on passively sensed digital trace data.”

R2.2 comment: *“My main concern is this of the not so detailed GT data collection methodology. Though it is obvious that the authors know the topic well, the presentation and reporting should be enhanced. That is, it should be presented like a step-by-step procedure that can be replicated by anyone. For example, period, category, region, quotes of not for more than one-word keywords, etc., are details that need to be included. In order to see how to structure and report a detailed GT methodology, please see the following two papers:*

a) <https://www.jmir.org/2021/7/e27044/>

b) <https://www.jmir.org/2020/8/e19611/>

(Note: I am not in the list of authors of any of these articles, nor am I in anyway requesting that you cite these documents (they are totally irrelevant with your work anyway); I am just including them in order to show how this comment could be addressed).”

R2.2 response: Thank you for these pointers. It was helpful to see how Google Trends data collection is described in these two very relevant publications.

We retrieved country-level Google Trends search interest time series from 1 January 2019 to 31 December 2020 for 18 countries: Brazil, Canada, Mexico, United States, France, Germany, Italy, Spain, United Kingdom, India, Indonesia, Japan, Egypt, Kenya, Nigeria, Australia, Sweden, Denmark (increased from 12 to 18 countries in response to Reviewer 3's comment; see below). The time series were collected at weekly granularity, for 1,432 food entities and 16 entities related to modes of accessing food. The time series were collected and calibrated using the Google Trends Anchor Bank library (<https://pypi.org/project/gtab/>). Since the complete list of queried entities is too long to be stated in the manuscript, it is available in our data repository (<https://github.com/epfl-dlab/foodle-trends>), in the "food_timeseries" and "modes_timeseries" files.

The details necessary to reproduce our data collection are now stated in Section 4.2 ("Search interest data collection"; page 10). Please note that, as described in the Results section (line 127) and in the Methods section (line 440), we query Freebase identifiers, which aggregate multiple surface forms and keywords. We did not perform any translation. Also note that we obtained time series at the food-category and access-mode levels only after data collection, by adding up time series of the respective individual entities in a post-processing step. We have now explained this more clearly (Section 4.2). The selection of the entities is described in Section 4.1.

R2.3 comment: *"Since GT data are normalized, the retrieved time series should exactly match the respective one for which non Google data (e.g., in this case mobility data) are used. I am certain that the authors have taken this into account and have adjusted the time series accordingly, but this should be explicitly mentioned in the Methods section, i.e., the authors should include the exact time frame for which they retrieved GT data, as well as when they retrieved said data, as the time of retrieval may slightly affect the downloaded time series."*

R2.3 response: As mentioned above (under "R2.2 response"), the Google Trends search interest time series were collected for the period from 1 January 2019 to 31 December 2020 for 18 countries. The dataset was collected between 9 February 2021 and 26 March 2021 (Australia, Brazil, UK, USA, Italy, India, France, Spain, Denmark, Mexico, Canada, Germany), and between 25 October 2021 and 2 November 2021 (Nigeria, Kenya, Indonesia, Egypt, Sweden, Japan). (The latter six countries were collected and added to the revision in response to Reviewer 3's comment; see below.) We have also stated these details in the Reporting summary.

The mobility time series were downloaded (for the 18 countries) for the period from February through December 2020. These two periods (spanned by the Google mobility data and the Google search interest data, respectively) are not exactly overlapping, since Google published COVID-19 mobility reports that begin only in February 2020. This is not an issue for our core analyses, however, since we detect the date when mobility drastically decreased and use only that date as the cut-off point in our search interest modeling. The only place where mobility and search interest time series are directly correlated is Section 2.2 (line 178, Figure 4 and Table 2). This does not pose a problem for our analysis, though, as the partial overlap is taken into account by correlating the mobility and search interest time series within the overlapping period only. We have now clarified this in Section 2.2 (line 182).

R2.4 comment: *“I understand (and support) the selection of countries that are diverse enough so that the effect can be studied from different angles, but possibly the study period for each country (or at least group of countries) should vary, since not all countries were equally affected in all waves.”*

R2.4 response: To keep our analyses consistent and comparable, we opted for studying the period from the beginning of 2019 until the end of 2020, across all countries.

Please note that our study design and our modeling approach already allow the modeled periods to vary between countries. Our main analyses based on RDD modeling take into account the same number of weeks before and after the shock of the mobility decrease, regardless of when that shock happened. The quadratic model is flexible enough to capture the temporal evolution of search interest across countries, even though the countries were not equally affected and the date of the shock itself varies across countries. This is illustrated in Figure S5: time is relative to the mobility decrease, and the moment $t = 0$ does not occur simultaneously in all studied countries.

Finally, although we kept the set of studied countries constant, we took advantage of the fact that not all countries were equally affected in our analysis of the dose-response relationship between mobility and search interest (Figure 5b).

R2.5 comment: *“I think that black should be replaced with another (lighter) color (e.g., gray) in Figure 5. Imagine the ink waste if this was printed!”*

R2.5 response: Thank you for pointing this out. We have replaced black with gray in the previous Figure 5 (now Figure 7).

R2.6 comment: *“Worldwide heat maps depicting the online interest in some (or even all, like one panel figure) of the examined terms would be a plus”*

R2.6 response: Thank you for this suggestion! We have now included an additional analysis (Figure 1) where we study how the interest in two specific entities (“recipe” and “restaurant”) changed in 2020 compared to 2019, worldwide. It is clearly visible that there was indeed a global increase in interest in recipes, and a global decrease in interest in restaurants. Even though this analysis is not as in-depth as the main modeling studies performed in the paper, we believe that it helps put the findings into perspective and argue that the findings are expected to generalize to some extent beyond the 18 countries on which we focus.

Reviewer #3

R3.1 comment: *“The study presents data on how life alterations related to the first wave of the covid-19 pandemic have affected interest in different dietary items at the population level. Methodologically, the authors used food-related Google Search volumes from 12 countries in the period 2019-2020. Changes to the volume and the nature of the food-related searches were visible in all countries and could be linked to the extent of mobility restrictions in the individual countries.*

This is an interesting and obviously topical paper. The results are clear and well-interpreted (for the most part) within the conditions tested. The paper is methodologically valuable as the long time window considered makes it possible to provide interesting insights into how persistent these changes are over time and how long it takes for things to “go back to normal”. This is a major advance compared to the existing literature on the topic which is mostly based on self-reports providing snapshots of much shorter time ranges.

My only significant criticism to the paper would be to the geographical distribution of the included countries. The authors state (LL. 110-112) that the 12 countries were chosen to achieve a diversity of geographical location but clearly there is an overrepresentation of western countries; in particular there is only one Asian country and no African country at all. This lack of representation clearly diminishes the global relevance of the paper, especially considering that almost 80% of the world population lives in Asia and Africa. Therefore, I would strongly encourage the authors to include additional countries from those regions..”

R3.1 response: Thank you for the encouraging comments. We are grateful for your comment regarding geographical coverage and for pushing us to improve the manuscript. We agree that the lack of representation of Asia and Africa diminished the global relevance of our work. We hence included six additional countries: five large Asian and African countries (Indonesia, Japan, Egypt, Kenya, and Nigeria), plus Sweden (see below). The 18 selected countries are now based on transparent predetermined criteria. We have accordingly revised all analyses and interpretations. We believe that doing so has substantially improved the manuscript.

The revised set of 18 studied countries was selected according to the following criteria (described on line 118, page 3): We selected the countries with the largest number of Internet users across continents, relying on statistics about the number of Internet users per country.¹ The People's Republic of China and Russia were not chosen from the set of Asian countries as listed by the resource we consulted since Google is not the most-used search engine there (Yandex and Baidu are; Google is banned from the People's Republic of China).

The revised set of studied countries is the following:

Canada, Mexico, United States (North America);
Brazil (South America);
France, Germany, Italy, Spain, United Kingdom (Europe);
India, Indonesia, Japan (Asia);
Egypt, Kenya, Nigeria (Africa);
Australia.

Additionally, to achieve a varying severity of lockdowns, Sweden and Denmark were added as contrasting cases, due to particularly mild COVID-19-induced mobility changes.

We believe that expanding the geographical coverage has strengthened our manuscript for the following reasons:

- (a) The main conclusions remain unchanged after incorporating the additional countries.
- (b) The Christmas comparisons became more meaningful when contrasted with countries with a non-Christian majority, where there are no regular seasonal peaks in food interest.
- (c) The absence of population-wide changes in Sweden and Japan, the countries with the least stringent public health interventions (together with Denmark, which was already among the 12 original countries), serves as further validation that there is a link between how the pandemic unfolded during 2020 and shifting dietary interests.

¹ <https://www.statista.com/markets/424/topic/537/demographics-use>

Finally, we have also performed an additional, truly global analysis (map visualizations in Figure 1), where we study how the interest in two specific search terms (“recipe” and “restaurant”) changed in 2020 compared to 2019, worldwide. As seen in the maps, there was indeed a global increase in interest in recipes, and a global decrease in interest in restaurants. Even though this analysis is not as in-depth as the main modeling studies performed in the paper, we believe that it helps put the findings into perspective and argue that the findings are expected to generalize to some extent to other regions.

Note that we originally selected the *globally* most popular dish entities (page 10, line 474). Therefore, the studied set of food entities already includes global dishes (such as Asian dishes), and therefore did not have to be changed when adding new countries.

R3.2 comment: *“A smaller issue was with the introduction, which I found, at times, insufficiently rooted in the existing literature. For example, at LL. 70-72 the authors state that a limit of existing literature is that it has focused on specific countries. This is not really true as there are already several international comparative studies...*

(such as e.g.

Molina-Montes et al. (2021). Impact of COVID-19 confinement on eating behaviours across 16 European countries: The COVIDiet cross-national study. Food Quality and Preference, 93, 104231.

Ismail et al. (2021). Assessment of eating habits and lifestyle during the coronavirus 2019 pandemic in the Middle East and North Africa region: a cross-sectional study. British Journal of Nutrition, 126, 757-766.

Ammar et al. (2020). Effects of COVID-19 home confinement on eating behaviour and physical activity: results of the ECLB-COVID19 international online survey. Nutrients, 12(6), 1583. – This is already included in the reference list)

... that have already established that there are differences between countries in the way dietary habits have been affected by covid-19 and even linked these differences explicitly to lockdown severity, which is one of the main finding of this paper. These studies does not the paper any less valuable, as the topic is still highly relevant and there are issues with previous studies (e.g., reliance on self-reported measures via surveys).”

R3.2 response: We are thankful for these highly relevant references. It is indeed true that there are several published comparative studies. We have hence edited the Introduction (line 76) to reflect this more truthfully.

We were not familiar with the two cross-national studies (Molina-Montes et al. and Ismail et al.), and added these references in the Introduction where we discuss the emerging behaviors observed in previous studies, such as cooking more frequently and weight gain (lines 59 and 67).

R3.3 comment: *“Another issue that I miss in the introduction is the topic of individual differences within countries. It is well established by now that the effect of the lockdown and other restrictive measures has been different for different segments of the population defined by (sex, age, BMI, pre-existing dietary habits, etc.). This point is not directly related to the author’s study which deals with population-level data, but I think it should at least briefly acknowledged: there is substantial literature on the topic and it is disingenuous to let readers with the impression that the effect of covid measures on people’s dietary interest are only (or even primarily) related to which country they live in.”*

R3.3 response: Thank you for raising this important point. We have now stated explicitly (line 48) that, although the population-scale measures taken to prevent the spread of the virus vary between countries, the implemented interventions disparately impact population segments within a country, depending on people’s demographics, health, and habits. We have also added references to the aforementioned studies (R3.2) that considered the impact of these factors.

R3.4 comment: *“LL. 144-146. Should a reference be provided for this statement?”*

R3.4 response: We incorporated references [15,43,65,93] to back up our statements regarding the severity of the interventions mandated in a country.

R3.5 comment: *“LL. 220-224. The measure of lockdown severity used by the authors makes very good sense but I see no reference so I assume it is an original approach; if so, it might be relevant to check whether it correlates with more formal ways of quantifying lockdown severity, e.g. using data from the Oxford COVID-19 Government Response Tracker (OxCGRT).”*

R3.5 response: The approach of relying on mobility reports published by Google and the subsequent estimation of mobility changepoints and lockdown severity based on the increase of time spent at home were adapted from ref. 71. We indeed failed to cite that reference when explaining how the severity of lockdown was determined. For clarity, we have now added the corresponding reference (line 238), describing the analyses presented in Figure 5.

Throughout this work, we chose to rely on mobility data, and not the official start of lockdown dates and government stringency statistics, since population-scale mobility

data is the closest indicator of the time spent at home. Regarding government responses, it is not guaranteed that measures would impact movement patterns across different countries homogeneously (people may have stayed more at home even before the lockdown was enacted, and people may stay at home more even where there might be no official lockdowns at all).

Nonetheless, we have further investigated this aspect, and have found that the lockdown severity as determined via mobility reports indeed correlates strongly with the Government Response Stringency Index (Pearson correlation 0.63 across the 18 studied countries, see table below). Furthermore, the mobility change points coincide with the implementation of measures, as demonstrated before [71].

	Percentage of time spent at home at the peak of reduced mobility	Government Response Stringency Index as of Apr 15, 2020*
Australia	21.71	77.78
Brazil	20.86	74.54
Canada	23.29	72.69
Denmark	16.86	68.52
Egypt	19	84.26
France	30.43	87.96
Germany	25.71	76.85
India	29.14	100
Indonesia	19.71	71.76
Italy	31.57	93.52
Japan	21.43	45.37
Kenya	24.57	88.89
Mexico	21.14	82.41
Nigeria	28	82.87
Spain	31.43	85.19
Sweden	10.14	64.81
UK	25.71	79.63
US	19.43	72.69

*Downloaded from:

<https://ourworldindata.org/grapher/covid-stringency-index?tab=table&time=2020-04-15..2020-08-01>

R3.6 comment: *“LL. 331-334. This link to stress, boredom, emotional eating etc., should also be discussed upfront (i.e., in the introduction) as it is often brought up in the literature. Here, I would rather discuss how the authors’ results are consistent with this view; for example Fig. 4 shows increases in interest for pastries but no increase in alcoholic beverages, so it is not clear that all results align with the “emotional eating” explanation.”*

R3.6 response: We have now rephrased the Introduction to explicitly explain (line 57) how concerns about the long-term implications of stress and boredom associated with the lockdown and subsequent emotional eating are suspected to be linked with both alcohol misuse and weight gain.

Regarding our findings and how they relate to these phenomena, it is indeed true that the absence of an increase in interest in alcoholic beverages is not consistent with the emotional eating explanation. We have added more discussion acknowledging this fact (line 379). While, at a population level, more interest in alcohol is associated with more time spent outside of home, there might be specific subpopulations of users who might still be susceptible to alcohol misuse, consistent with the emotional consumption hypothesis.

R3.7 comment: *“LL. 369-380. This part could be shortened a bit. Also, it sounds more defensive that it needs to be. It is certainly true that searching for a food is not the same as eating food, but you can say the same about the existing literature which is largely based on surveys. A well-known fact in nutrition research is that self-reported data correlate quite poorly with actual food intake and you also have issues such as social desirability bias that may lead people to over(under) report certain items. At least this paper is based on actual behaviors so in fact I would actually regard it as having a higher validity than most existing studies on this topic.”*

R3.7 response: Thank you for pointing this out. We fully agree and consider the limitations and shortcomings of self-reported data in the Introduction when motivating the study. In response, to avoid coming across as overly defensive, we have condensed the corresponding Discussion paragraphs (line 394), while remaining transparent about the most important limitations of our study.

R3.8 comment: *“Table 1. Two minor issues: 1) Pasta does not generally contain eggs. 2) The description of the pie category (“baked dish”) should be improved.”*

R3.8 response: Thank you for helping us make the category descriptions more precise (Table 1). We have removed the eggs from the “pasta, pizza, and noodle” category description since pasta and noodles often indeed do not contain eggs. We have also updated the “pie” category description, now stating that it refers to “baked dishes usually made of a pastry dough casing, containing a filling of various sweet or savory ingredients”.

Reviewers' Comments:

Reviewer #2:

Remarks to the Author:

All comments from the first review round have been adequately addressed.

Reviewer #3:

Remarks to the Author:

Population-scale dietary interests during the COVID-19 pandemic (NCOMMS-21-38571A)

The authors have done an excellent job in addressing my comments to the original submission. Thank you for accommodating the request for additional geographical coverage. I think it has improved the paper. The inclusion of Sweden and the DK/SE comparison made it for a very interesting side note regarding the effect of the lockdown severity.

My other comments were mostly minor and pertained to qualifying different statements and interpretation, all of which have been adequately addressed. Honestly, the authors' response document looked a masterclass in how authors should reply to reviewers' comments.

I have no further comments am happy to recommend the paper for acceptance in its current form.

Response to reviewers

Reviewer #2 (Remarks to the Author):

All comments from the first review round have been adequately addressed.

Response: We thank you again for helping us improve this manuscript.

Reviewer #3 (Remarks to the Author):

The authors have done an excellent job in addressing my comments to the original submission. Thank you for accommodating the request for additional geographical coverage. I think it has improved the paper. The inclusion of Sweden and the DK/SE comparison made it for a very interesting side note regarding the effect of the lockdown severity.

My other comments were mostly minor and pertained to qualifying different statements and interpretation, all of which have been adequately addressed. Honestly, the authors' response document looked a masterclass in how authors should reply to reviewers' comments.

I have no further comments am happy to recommend the paper for acceptance in its current form.

Response: Thanks for helping us improve this manuscript with constructive and thoughtful feedback that led to a more comprehensive study!